# ONLINE SPECULATIVE DECODING

## ABSTRACT

Speculative decoding is a pivotal technique to accelerate the inference of large language models (LLMs) by employing a smaller draft model to predict the target model's outputs. However, its efficacy can be limited due to the low predictive accuracy of the draft model, particularly when faced with diverse text inputs and a significant capability gap between the draft and target models. We introduce online speculative decoding to address this challenge. The main idea is to continually update (multiple) draft model(s) on observed user query data using the abundant excess computational power in an LLM serving cluster. Given that LLM inference is memory-bounded, the surplus computational power in a typical LLM serving cluster can be repurposed for online retraining of draft models, thereby making the training cost-neutral. Since the query distribution of an LLM service is relatively simple, retraining on query distribution enables the draft model to more accurately predict the target model's outputs, particularly on data originating from query distributions. As the draft model evolves online, it aligns with the query distribution in real time, mitigating distribution shifts. We develop a prototype of online speculative decoding based on online knowledge distillation and evaluate it using both synthetic and real query data on several popular LLMs. The results show a substantial increase in the token acceptance rate by 0.1 to 0.65, which translates into $1.22\times$ to $3.06\times$ latency reduction.

## 1 INTRODUCTION

Large language models (LLMs) such as GPT-4 (OpenAI, 2023), Claude (Bai et al., 2022), and Llama (Touvron et al., 2023a;b) are rapidly reinventing today's applications. Many companies are racing to deploy LLMs in their vertical domains, such as search, chatbots, and virtual assistants. Since most of these applications demand low latency, optimizing LLM serving latency is of vital importance and can directly translate into better quality of service and cost reduction.

The latency of today's LLM service is unfortunately very high. This is primarily because serving a user query requires multiple serial evaluations of the LLM, each generating only one token of the response. An emerging solution to reduce the latency is speculative decoding. Speculative decoding employs a smaller model to speculate multiple output tokens of the target (large) model, then lets the target LLM verify these speculations in parallel. Then, if the verification of a token fails, the large model must recompute from that point. Therefore, the performance of speculative decoding primarily depends on the speculation accuracy of the small model. In the presence of diverse text inputs, the accuracy of existing speculative decoding methods is unfortunately not very high, due to the capability gap between the draft and target model. Employing a larger, more accurate model however defeats the purpose of speculative decoding as it potentially increases latency.

To address this challenge, we introduce a novel method, *online speculative decoding*, specifically designed for online LLM services. The method leverages the abundant redundant compute, termed as "spare flops," available in a typical LLM serving cluster to continuously retrain (multiple) small draft models through online learning on query data posted to the LLM service. Our approach is simple and offers several significant advantages. First, user queries to a specific LLM service often exhibit a common domain-specific distribution (Zheng et al., 2023a), reflecting shared usage patterns. While accurately speculating the larger model's outputs on *any diverse input* is challenging, it is feasible to enhance the draft model's prediction accuracy, *only for similar inputs posted to the service*, characterized by the query distribution. This can be achieved by finetuning the draft model on user query distribution or finetuning multiple draft models, each on a cluster of the query distribu-

tion, and selecting the appropriately specialized draft model to speculate based on the class of inputs they are trained on. As shown in §5.2, we show that it is possible to train multiple draft models, each for a different language or topic. Second, the primary bottleneck for transformer-based LLM inference is the accelerator's memory bandwidth, as generating each word requires loading the model weights from HBM to SRAM as well as reading the KV cache on all previous words. This results in a substantial amount of unused compute, especially during non-spike traffic hours (Spector & Re, 2023; Chen et al., 2023; Kwon et al., 2023), in an LLM serving cluster. We demonstrate that these spare FLOPs can be effectively repurposed for online retraining of draft models, with inconspicuous retraining cost (§4.2.2). Third, since tuning is performed online, the draft models continuously evolve over time based on the observed query data, which ensures high speculation accuracy even when faced with shifts in query distribution.

Based on these insights, we develop an online speculative decoding framework to improve the efficiency of online LLM serving. To align the draft model with the target model on a newly observed user query, we develop a new online learning algorithm based on Generalized Knowledge Distillation (GKD) (Gu et al., 2023; Agarwal et al., 2023). The algorithm keeps track of the recent queries that the draft model has speculated incorrectly, and forces the draft model to emulate the target model's outputs on these queries. The algorithm performs GKD-based gradient update opportunistically only when spare flops are available, hiding the overhead.

In summary, this paper makes the following contributions:

- We introduce online speculative decoding to reduce LLM serving latency by adapting (multiple) draft models on the fly using query data and knowledge distillation.
- We explore various GKD methods for constructing draft models and identify the most effective variants, suggesting them as superior alternatives to existing finetuning methods in offline settings.
- Our method demonstrates a significant improvement in token acceptance rate by 10-65% on diverse datasets, translating to 1.2-3.1× reduction in latency theoretically, with a negligible additional cost. It surpasses existing methods which construct static draft models using fine-tuning or distillation on offline datasets, and matches the hypothetical accuracy achieved if all query data were available a priori.

## 2 RELATED WORK

LLMs have become pervasive in today's AI applications, underscoring the importance of optimizing LLM inference. Numerous system optimizations have been developed to optimize the throughput of LLM serving (Yu et al., 2022; Kwon et al., 2023). This paper particularly concentrates on a significant strand of research, speculative decoding, aimed at reducing the latency of LLM inference.

**Speculative decoding.** Speculative decoding (Leviathan et al., 2023; Chen et al., 2023) accelerates LLM decoding by employing a (small) draft model to predict the outputs of the larger target model, which are then verified by the target model. Typically, the draft model, while having fewer parameters, is pretrained using the same training data as the target mode, resulting in a negotiable inference cost but with compromised capability. If the draft model can correctly predict more than one token per verification step, the memory I/O for accessing the model weights and KV cache at inference is amortized across multiple output tokens, thereby reduces latency, especially since LLM inference is often constrained by GPU HBM bandwidth. The efficacy of speculative decoding largely hinges on the draft model's ability to accurately predict the target model's outputs. Existing work improves the speculation accuracy by using multiple collectively boosted (Miao et al., 2023) or staged (Spector & Re, 2023) draft models, or retraining the target model with auxiliary prediction heads as a draft model (Cai et al., 2023; Stern et al., 2018). These methods predominantly assume a static draft model post-deployment. In contrast, our work introduces a framework that actively adapts the draft model to the evolving user query distribution on the fly, irrespective of the draft model's construction.

**Distillation for auto-regressive models.** Knowledge distillation (KD) is a framework to generate smaller models that emulate the performance of larger models. However, KD in its conventional form has been observed to be less effective for LLMs. Gu et al. (2023) extend KD to autoregressive LLMs by decoding from the student model and optimizing the reserve KL divergence between students and teachers. Further, Agarwal et al. (2023) introduce generalized knowledge distillation

(GKD) to optimize a linear combination of the forward KL and reverse KL between teacher and student, using a blend of teacher- and student-sampled data. Drawing inspiration from both works, our paper applies KD to speculative decoding for LLMs. We empirically determine the most effective KD variant for maximizing the draft model's accuracy, and extend it to dynamically generate draft models for online LLM services.

## 3 BACKGROUND

We first briefly review speculative decoding (Leviathan et al., 2023), a critical technique that accelerates inference of a large target LLM $p(\cdot|\boldsymbol{x})$ with token proposals from a small draft model $q_{\boldsymbol{\theta}}(\cdot|\boldsymbol{x})$. $\boldsymbol{x}$ denotes the concatenation of the input prompt and already generated tokens. The two distributions are both auto-regressive. We emphasize the parameters $\boldsymbol{\theta}$ of the draft model because we usually need to tailor them according to the target LLM for more substantial acceleration.

Speculative decoding uses a (small) draft model to propose $k$ tokens $\boldsymbol{y} \triangleq \{y_i\}_{i=1}^k \sim q_{\boldsymbol{\theta}}(\cdot|\boldsymbol{x})$, and let the target LLM estimate the $k+1$ probabilities, $\{p(y|\boldsymbol{x}, \boldsymbol{y}_{<i})\}_{i=1}^{k+1}$[1], in parallel. With $i$ rising from 1 to $k$, speculative decoding accepts the proposal $y_i$ if $u \leq p(y_i|\boldsymbol{x}, \boldsymbol{y}_{<i})/q_{\boldsymbol{\theta}}(y_i|\boldsymbol{x}, \boldsymbol{y}_{<i})$ where $u \sim U[0,1]$; otherwise exits. Let $a$ denote the number of accepted tokens, which takes values in $\{0, \dots, k\}$. We can sample an additional token $y_{a+1}$ from the following distribution

$$p'(y) = \begin{cases} p(y|\boldsymbol{x}, \boldsymbol{y}_{<a+1}) & \text{if } a = k \\ \text{norm}(\max(0, p(y|\boldsymbol{x}, \boldsymbol{y}_{<a+1}) - q_{\boldsymbol{\theta}}(y|\boldsymbol{x}, \boldsymbol{y}_{<a+1}))) & \text{otherwise} \end{cases} \quad (1)$$

where $\text{norm}(\cdot)$ makes the probabilities over the vocabulary sum to 1.

Prior work has shown that the resulting samples $\tilde{\boldsymbol{y}} \triangleq \{y_1, \dots, y_{a+1}\}$ strictly follow the distribution of the target LLM $p(\cdot|\boldsymbol{x})$ (Leviathan et al., 2023). We concatenate $\tilde{\boldsymbol{y}}$ to $\boldsymbol{x}$ and repeat the above process until meeting $\langle \text{EOS} \rangle$. Each run of the target LLM generates $a+1$ tokens with $a \geq 0$. This ensures that at least one new token is generated even in the worst case. The generation process can be significantly accelerated if the draft LLM better approximates the target one, particularly $a$ is larger for each target LLM run.

**Expected acceptance rate & speedup.** The acceptance rate, denoted as $\alpha$, serves as a measure of how closely the draft model approximates the target model. It is defined as the expected probability that speculative decoding will accept a proposal token given the prompt $y_i \sim q_{\boldsymbol{\theta}}(y_i|\boldsymbol{x}, \boldsymbol{y}_{<i})$. This rate directly influences the expected length ($\mathbb{E}(|\tilde{\boldsymbol{y}}|)$) of $\tilde{\boldsymbol{y}}$ for each target LLM run and the speedup brought by speculative decoding.

Assuming that the $k+1$ simultaneous evaluations of the target LLM $p$ take roughly the same amount of time as generating a single token in parallel, let $c$ be the time ratio for a single run between $q_{\boldsymbol{\theta}}$ and $p$. The expected generation length of a single target LLM run and the speedup in the total wall time due to speculative decoding is represented as (Leviathan et al., 2023):

$$\mathbb{E}(|\tilde{\boldsymbol{y}}|) = \frac{1 - \alpha^{k+1}}{1 - \alpha}, \quad \mathbb{E}(speedup) = \frac{1 - \alpha^{k+1}}{(1 - \alpha)(kc + 1)}. \quad (2)$$

We depict the speedup for varying values of $\alpha$ in Figure 1, which demonstrates the importance of $\alpha$ in affecting the speedup.

**Observation.** Interestingly, we can actually enhance $\alpha$ based on a key observation: the speculative decoding process inherently identifies the inaccuracies of the small draft LLM and offers correct solutions for these inaccuracies. This essentially means that we receive valuable insights on the areas and strategies to refine the draft model at no additional cost. Viewed through the lens of online learning, we can effortlessly accumulate a set of input-output pairs, denoted as $([\boldsymbol{x}, \boldsymbol{y}_{<a+1}], p(y|\boldsymbol{x}, \boldsymbol{y}_{<a+1}))$, that have yet to be assimilated by the draft LLM, paving the way for its subsequent optimization. Given the reduced size of the draft model (for instance, it may be over $20\times$ smaller than the target model), its tuning is not only efficient but also viable for real-time online adjustments. Prior work (Leviathan et al., 2023; Miao et al., 2023) has primarily approached speculative decoding in an offline manner, meaning the draft model remains static during online deployment. We next develop online speculative decoding to bridge this gap.

---

[1] $\boldsymbol{y}_{<i}$ refers to $\{y_j\}_{j=1}^{i-1}$.

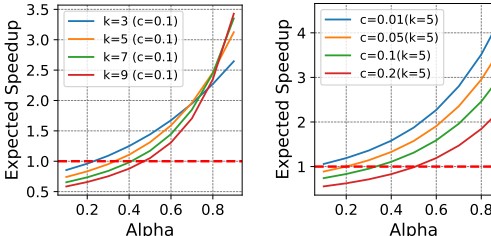

Figure 1: Speculative decoding speedups for varying values of $\alpha$ in Figure 1. For smaller $\alpha$ values, speculative decoding may even degrade performance (indicated by a speedup $< 1$), particularly when the draft model is sizeable. Furthermore, the relationship between speedup and $\alpha$ is superlinear; doubling the acceptance rate can yield a speedup exceeding $2\times$.

# 4 ONLINE SPECULATIVE DECODING

We propose the online speculative decoding approach to update the draft model dynamically for more effective suggestions. We frame the learning problem based on the aforementioned auxiliary information as online knowledge distillation, where the teacher and student models correspond to the target and draft LLMs in speculative decoding, respectively. We elaborate on the details below.

## 4.1 KNOWLEDGE DISTILLATION FOR SPECULATIVE DECODING

Knowledge distillation is a general framework to align the predictive distribution of a small model (i.e., student model) with that of a larger one (i.e., teacher model). Prior research has utilized knowledge distillation to compress neural networks, resulting in decreased inference costs and memory requirements. We posit that knowledge distillation is highly effective for speculative decoding. In this approach, the draft model acts as the student and the target model serves as the teacher. During speculative decoding, we possess complete information on both the proposed and verified probabilities of each token. This information helps to construct objectives for distilling the draft model, aligning its output distributions with those of the target model and thereby improving the token acceptance rate of the draft model. The distillation loss generally takes the form of:

$$\ell(\boldsymbol{\theta}) = \frac{1}{n_B} \sum_{\boldsymbol{x}^{(i)} \in \mathcal{B}} \ell(\boldsymbol{x}^{(i)}, \boldsymbol{\theta}), \quad \ell(\boldsymbol{x}, \boldsymbol{\theta}) = D(p(\cdot|\boldsymbol{x}) \| q_{\boldsymbol{\theta}}(\cdot|\boldsymbol{x})), \tag{3}$$

where $\mathcal{B} = \{\boldsymbol{x}^{(i)}\}_{i=1}^{n_B}$ denotes a batch of inputs and $D$ denotes some distance measure.

**Distance measure.** In the case of auto-regressive models, the prediction distribution is categorical at each token. Often, we can augment the predicted logits with a tunable temperature $\tau$ for softmax transformation. We then use the popular forward KL and reverse KL (RKL), as well as their mixture (i.e., the JSD divergence) to instantiate $D$ (Agarwal et al., 2023; Gu et al., 2023):

$$\begin{aligned}
\ell_{KL}(\boldsymbol{x}, \boldsymbol{\theta}) &= D_{\mathrm{KL}}(p(\cdot|\boldsymbol{x}) \| q_{\boldsymbol{\theta}}(\cdot|\boldsymbol{x})), \\
\ell_{RKL}(\boldsymbol{x}, \boldsymbol{\theta}) &= D_{\mathrm{KL}}(q_{\boldsymbol{\theta}}(\cdot|\boldsymbol{x}) \| p(\cdot|\boldsymbol{x})), \\
\ell_{JSD[\beta]}(\boldsymbol{x}, \boldsymbol{\theta}) &= \beta D_{\mathrm{KL}}\left(p(\cdot|\boldsymbol{x}) \big\| p_{\boldsymbol{\theta}}^{\beta}(\cdot|\boldsymbol{x})\right) + (1-\beta) D_{\mathrm{KL}}\left(q_{\boldsymbol{\theta}}(\cdot|\boldsymbol{x}) \big\| p_{\boldsymbol{\theta}}^{\beta}(\cdot|\boldsymbol{x})\right),
\end{aligned} \tag{4}$$

where $p_{\boldsymbol{\theta}}^{\beta}(\cdot|\boldsymbol{x}) \triangleq \beta p(\cdot|\boldsymbol{x}) + (1-\beta) q_{\boldsymbol{\theta}}(\cdot|\boldsymbol{x})$. These objectives diverge from the conventionally used label-based fine-tuning objectives in speculative decoding, as highlighted in (Miao et al., 2023; Leviathan et al., 2023). As shown in Section 5.1, objectives based on the KL divergence prove to be more effective. This is because distributions convey richer information than mere labels, thereby enhancing their capability to guide the student model (Hinton et al., 2015). Additionally, these objectives enhance convergence rates (He et al., 2022) and bolster calibration. The reverse KL is highlighted for its mode-seeking behavior, offering unique advantages (Gu et al., 2023). In our study, and in alignment with previous research (Agarwal et al., 2023), we empirically determine that the optimal distance measure can vary depending on the tasks and the relative capacities of the teacher and student models (see §5.1).

---

**Algorithm 1** Online Speculative Decoding.

---

1: **Input:** Target LLM $p(\cdot|\boldsymbol{x})$, draft LLM $q_{\boldsymbol{\theta}}(\cdot|\boldsymbol{x})$, warmup dataset $\mathcal{D}$, online data stream $\mathcal{S}$, guess number $k$, temporary buffer $\mathcal{R}$, replay buffer $\mathcal{Q}$, update interval for the draft model $I$.
2: Pre-train $q_{\boldsymbol{\theta}}$ to approximate $p$ with data from $\mathcal{D}$ by minimizing $\ell(\boldsymbol{x}, \boldsymbol{\theta})$ using Equation (5);
3: $i \leftarrow 0$;
4: $\mathcal{Q} \leftarrow []$;
5: $cur\_len = |\boldsymbol{x}|$ // Total sequence length, including prompt length and tokens generated so far.
6: **while** True **do**
7:     $\mathcal{R} \leftarrow []$ // List of ($error\_index$, target logits at $error\_index$) pairs for a single request.
8:     $\boldsymbol{x} \sim \mathcal{S}, i \leftarrow i + 1$;
9:     **while** $\langle\text{EOS}\rangle$ not in $\boldsymbol{x}$ **do**
10:         $\boldsymbol{y} = \{y_1, ..., y_k\} \sim q_{\boldsymbol{\theta}}(\cdot|\boldsymbol{x})$;
11:         Estimate $\{p(y|\boldsymbol{x}, \boldsymbol{y}_{<i})\}_{i=1}^{k+1}$ in parallel;
12:         Determine number of accepted tokens $a$ and sample one more token, yielding $\boldsymbol{y} = \{y_1, \ldots, y_{a+1}\}$;
13:         $cur\_len \leftarrow cur\_len + a + 1$;
14:         $error\_index \leftarrow cur\_len$;
15:         Append $(error\_index, p(y|\boldsymbol{x}, \boldsymbol{y}_{<a+1}))$ to $\mathcal{R}$;
16:         $\boldsymbol{x} \leftarrow [\boldsymbol{x}, \boldsymbol{y}_{<a+2}]$;
17:     **end while**
18:     Append $(\boldsymbol{x}, \mathcal{R})$ to $\mathcal{Q}$;
19:     **if** $i \bmod I = 0$ **then**
20:         Update $q_{\boldsymbol{\theta}}$ on $\mathcal{Q}$ to minimize $\ell(\boldsymbol{x}, \boldsymbol{\theta})$ analytically;
21:         $\mathcal{Q} \leftarrow []$;
22:     **end if**
23: **end while**

---

**Sampling and gradient estimation.** Estimating the above objectives involves the expectation over $q_{\boldsymbol{\theta}}(\cdot|\boldsymbol{x})$ or $p(\cdot|\boldsymbol{x})$, which should be expanded recursively. Once the recursion depth exceeds 1, we can not analytically compute $D_{\text{KL}}$ but hinge on Monte Carlo approximation. When sampling from $q_{\boldsymbol{\theta}}(\cdot|\boldsymbol{x})$, we should differentiate through the sampling process for unbiased gradient estimation. However, this leads to policy gradient-style estimators and should rely on elaborate policies such as reward hacking and single-step regularization to reduce gradient variance and stabilize training (Gu et al., 2023).

In comparison, a more straightforward approach is to omit the differentiation through the sampling process (Agarwal et al., 2023), where the sample $\boldsymbol{y}$ is directly plugged into the objective:

$$\ell(\boldsymbol{x}, \boldsymbol{\theta}) \approx \sum_{j=1}^{|\boldsymbol{y}|+1} D(p(y|\boldsymbol{x}, \boldsymbol{y}_{<j}) \| q_{\boldsymbol{\theta}}(y|\boldsymbol{x}, \boldsymbol{y}_{<j})). \quad (5)$$

This way, various distance measures can be readily applied. Besides, the sampling becomes disentangled from the distance measure. i.e., we sample $\boldsymbol{y}$ from an arbitrary mixture of $p(\cdot|\boldsymbol{x})$ and $q_{\theta}(\cdot|\boldsymbol{x})$ but use KL, RKL or JSD for estimating the distribution mis-alignment.

Intuitively, the samples from the teacher model are usually coherent, which may raise difficulties in fitting the small student model, while samples from the student model may be less structured or even meaningless. A workaround strategy is to trade off between them via mixed sampling (Gu et al., 2023), i.e., $y_j \sim \beta p(\cdot|\boldsymbol{x}, \boldsymbol{y}_{<j}) + (1 - \beta)q_{\boldsymbol{\theta}}(\cdot|\boldsymbol{x}, \boldsymbol{y}_{<j})$.

## 4.2 ONLINE KNOWLEDGE DISTILLATION

This section expands the application of knowledge distillation for speculative decoding in online environments. The approach enables improving the performance of draft model using results from speculative decoding, thus dynamically adapting to the query distribution and improving token acceptance rate. We also discuss the trade-off of our approach when integrating LLM serving systems.

### 4.2.1 ALGORITHM

We depict our online speculative decoding algorithm (OSD) in Algorithm 1. OSD begins by training the draft model using the warmup dataset (Line 2). The serving system then continuously handles incoming requests (as described in Lines 6 to 23). For each request, it uses standard speculative

decoding (Lines 10-11) to generate responses until the ⟨EOS⟩ token. Concurrently, OSD tracks the token index ($error\_index$) and target logits where the draft model proposes the wrong tokens (Line 15). Leveraging tracked information, OSD updates the draft model every $I$ iteration, with $I$ being a dynamically adjustable parameter. OSD updates the draft model with different loss functions (Line 20) as described in Section 4.1. The choice of loss function depends on the specific (draft, target) model pairs and the corresponding input data.

**Discussion.** OSD utilizes a replay buffer, $\mathcal{Q}$, to capture all pertinent information for updating the draft model. Various eviction policies can be employed to maintain a compact size for $\mathcal{Q}$. For example, one could opt to retain only the most informative pairs or the most recent entries. Similarly, users have the option to retain data in $\mathcal{Q}$ even after utilizing it to update the model multiple times. Determining the optimal eviction/retention strategy is a subject for future exploration. In the current study, we refrain from evicting any pairs and release $\mathcal{Q}$ after each model update. Furthermore, $I$ is a dynamic parameter. Depending on the system load and the rate at which the query distribution changes, users can adjust $I$ accordingly. For example, we can perform a gradient update opportunistically only when the service traffic is not on spike (i.e., spare flops are available). Overall, OSD continuously improves the draft model's approximation (indicated by increased token acceptance rate $\alpha$) by learning from the target model during the serving phase. We next demonstrate how the enhanced acceptance rate directly contributes to a reduction in request latency.

### 4.2.2 LATENCY & FLOPS ANALYSIS

**Latency.** As detailed in Appendix A.2, compared with standard speculative decoding, the expected speedup for online speculative decoding is $\frac{1+\alpha_2+\alpha_2^2+...+\alpha_2^k}{1+\alpha_1+\alpha_1^2+...+\alpha_1^k}$. Based on the data from our experiment (refer to Table 1), when compared to standard speculative decoding, we expect a speedup improvement for Vicuna-7B (LLaMA-160M as the draft model) by factors of $2.42\times$, $1.43\times$, $1.64\times$, and $1.22\times$. Similarly, for Flan-T5-XL 3B (T5-small 80M as the draft model), the speedup enhancements are $3.06\times$, $1.76\times$, $2.72\times$, and $1.55\times$ across the four evaluated datasets.

**FLOPs.** (1) The FLOPs required to update the draft model are significantly fewer than those needed for inference on a large model. As elaborated in Appendix A.3, for the two evaluated model pairs, the FLOPs ratio between the target model and the draft model is 18.75 for the pair (LLaMA-160M, Vicuna7B), and 12.6 for the pair (T5-small 80M, Flan-T5-XL 3B). (2) In practical systems, the FLOPs required for inference are significantly below the machine's capacity. The Appendix A.3 provides an analysis of Arena chatbot traces where the cluster's computational utilization is under 1 percent. Given the above two observations, it becomes evident that the FLOPs spent on inference and updating the draft model are relatively insignificant when juxtaposed with the FLOPs consumed while operating the target model and the cluster's total FLOPs.

## 5 EXPERIMENTS

To assess the efficacy of our method, we initially evaluate its ability to improve the token acceptance rate ($\alpha$) within an offline context. This provides us with a theoretical upper bound on the performance improvements achievable when the query distribution remains constant. Subsequently, we examine the approach's impact in an online environment, discovering that the acceptance rate improves even with a moderate amount of data while maintaining accuracy levels comparable to those in the offline scenario. Throughout our experiments, we employ two target models ($M_p$): Vicuna-7B (Chiang et al., 2023) and FLAN-T5-XL (3B) (Chung et al., 2022). Specifically for Vicuna-7B, we utilize LLaMA-160m (Miao et al., 2023) as the draft model ($M_q$). For FLAN-T5-XL, we use T5-Small (Raffel et al., 2020) as the draft model. We evaluate performance across four diverse datasets: Text-to-SQL (Spider) (Yu et al., 2018), graduate school math (Gsm8k) (Cobbe et al., 2021), Python code generation (Code-search-Python) (Husain et al., 2019), and financial question answering (Alpaca-finance) (Bharti, 2023). In all experiments, we set the number of proposed tokens to 5 for speculative decoding. For all online experiments, we fix the update interval $I$ at 8.

Table 1: Token acceptance rates ($\alpha$) after two epochs. **FT**: Finetuning on teacher-generated labels. **TF, SF, MixF**: Teacher, student, and mix token sampling respectively, all with forward KL.

| Model | Task | Original | FT | TF | SF | MixF |
|---|---|---|---|---|---|---|
| Vicuna-7B | Spider | 0.28 | 0.74 | **0.76** | 0.62 | 0.70 |
| | Gsm8k | 0.58 | 0.74 | **0.75** | 0.67 | 0.73 |
| | Code-search-Python | 0.38 | **0.65** | **0.65** | 0.51 | 0.61 |
| | Alpaca-finance | 0.57 | **0.68** | 0.67 | 0.63 | 0.65 |
| FLAN T5-XL | Spider | 0.13 | 0.33 | **0.78** | 0.67 | 0.70 |
| | Gsm8k | 0.29 | 0.50 | **0.62** | 0.51 | 0.55 |
| | Code-search-Python | 0.28 | 0.44 | **0.81** | 0.67 | 0.78 |
| | Alpaca-finance | 0.39 | 0.56 | **0.63** | 0.59 | 0.60 |

## 5.1 OFFLINE EVALUATION

In this section, we assess the efficacy of employing knowledge distillation to train a small model specifically for speculation in an offline environment. In such a setting, the speculative $M_q$ model has unrestricted access to the dataset, and the query distribution remains stable. To emulate these offline conditions, we distill the $M_q$ using the training dataset for two epochs and subsequently evaluate its performance by measuring the average token acceptance rate ($\alpha$) on the test set. As detailed in Section 4.1, we evaluated various sampling methods, namely teacher sampling, student sampling, and mix token-level sampling. Table 1 displays the token acceptance rate of the draft model for each method, using forward KL as the distance metric on the test dataset. For comparison, we also provide the acceptance rate for teacher-generated label fine-tuning and the original model.

For both the Vicuna-7B and FLAN-T5-XL models, the teacher sampling method outperforms others by achieving the highest acceptance rate. Furthermore, knowledge distillation has proven its efficacy in enhancing the draft model's approximation, resulting in a high token acceptance rate. Lastly, we experimented with different distance measurements like reverse KL and JSD. Nevertheless, these measurements either paralleled or underperformed when compared to forward KL. The optimal distance measurement or sampling method varies depending on the task and model, and we leave it to future work to find the best combination.

## 5.2 ONLINE EVALUATION

**Online Learning.** First, we evaluate the effectiveness of our online algorithm by addressing two key questions: (1) Does the online algorithm increase the token acceptance rate? And is this enhancement comparable to the rates achieved in offline settings, which serve as an upper bound given their full access to data? (2) How quickly does the online algorithm increase the token acceptance rate, thereby indicating that the compact model has grasped the underlying distribution?

In our approach, we replicate the online serving process by iterating through the datasets, extracting prompts, and streaming generation requests. The system utilizes speculative decoding for each of these requests. Throughout this serving phase, we continually refine the speculative models, as detailed in Algorithm 1. For our baseline, we envision a scenario where the serving system has the capability to collect data offline in order to distill an initial draft model. This model is subsequently deployed online to cater to future requests. This process is simulated by using 10% of the dataset to distill the draft model, which remains static during online serving. For evaluation metrics, we calculate token acceptance rates averaged over the most recent 50 requests. This demonstrates $M_q$'s efficacy on the most current data.

As depicted in Figure 2, both for Vicuna-7B and FLAN-T5, in the beginning, OSD yields a lower token acceptance rate in comparison to the offline distilled model. Nevertheless, these acceptance rates rise swiftly as the draft model is exposed to more data. We also annotate the token acceptance rate from the offline setting to highlight the potential peak performance that the online serving system could reach. In all instances, the online context can achieve comparable results. In some scenarios, OSD even surpasses the token acceptance rate of the offline test alphas. This discrepancy can be attributed to the fact that offline test alphas are assessed on the entire test dataset, whereas the online alphas represent the moving average of the latest 50 requests. It's plausible that OSD per-

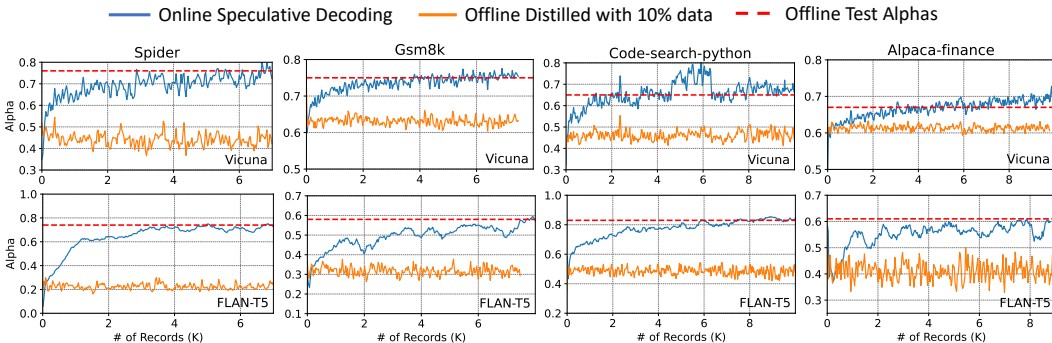

Figure 2: Online acceptance rate ($\alpha$) across different datasets. The x-axis represents the number of records that OSD has processed. Alpha is averaged over the most recent 50 records.

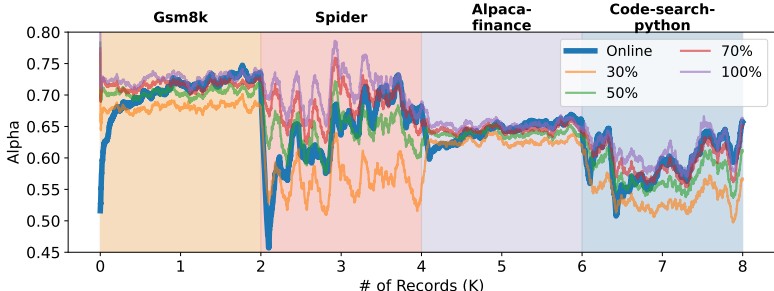

Figure 3: Distribution Shift: Alpha is averaged over the most recent 100 records.

forms optimally on specific data subsets, particularly if those subsets are more narrowly distributed than the complete dataset.

**Distribution Shifts.** We evaluate OSD's ability to adapt to changes in data distribution. We detail the experiment setting in Appendix A.6. As illustrated in Figure 3, OSD's alpha value dips notably at distribution boundaries, especially around 2K, 4K, and 6K records. This is anticipated since the draft model initially struggles when faced with a new distribution. However, the alpha value rebounds quickly as OSD processes more data, highlighting its adaptability to shifting query distributions.

We also compared our results to those from a static setting. To ensure the draft model wasn't just memorizing data, we chose samples distinct from the online evaluation data. These samples correspond to 30%, 50%, 70%, and 100% of each dataset's online evaluation volume, at 0.6K, 1K, 1.4K, and 2K quantities respectively. As depicted in Figure 3, upon an initial shift in query distribution, OSD's performance aligns with or slightly trails the distillation with 30% data. However, it quickly catches up, matching or even surpassing performances seen with 70% to 100% data access. This highlights OSD's ability to rival models fully exposed to the query distribution, even without intimate knowledge of the underlying query dynamics.

**Real Workloads.** We evaluate OSD on real LMSYS-chat conversations (Appendix A.7) that span 4 months. First, we categorize conversations based on the language and we focus on conversations among the top five languages, excluding English. For every chosen language, we use an independent LLaMA-160M to serve as our draft model. All draft models share the same Vicuna-7B as the target model. The token acceptance rate, averaged over the latest 100 requests, showed in Figure 4, reveals that OSD's enhances rates by 0.1 to 0.2, even with under 2K data points. Notably, Japanese was the easiest while Portuguese was the toughest. We also clustered English conversations by topics using the fine-tuned distilled Bert model (Luqmani, 2023), focusing on the top five. For topics with over 5K conversations, we sampled evenly to keep it within 5K. Figure 4 shows acceptance rates above 0.6 across topics, with Social and Computer discussions peaking near 0.8.

## 5.3 QUALITATIVE ANALYSIS

In this section, we conduct a comprehensive analysis to understand how our method enhances the token acceptance rate, and which tokens the draft model acquires across varying query distributions.

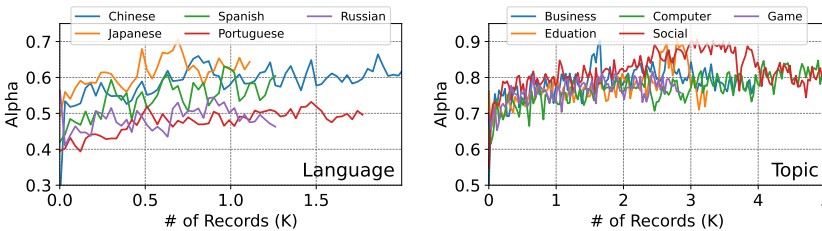

Figure 4: Chatbot Arena Conversations clustered by language and topic.

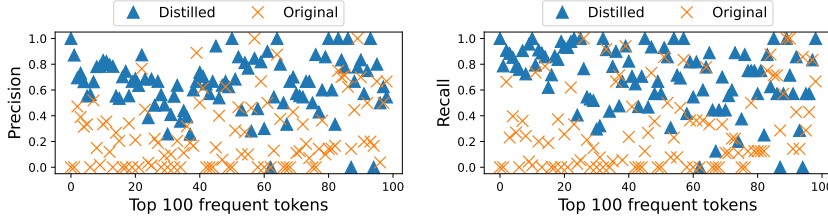

Figure 5: Precision and recall of high-frequency tokens. The x-axis shows the rating of the tokens based on their occurrence in the generated answers. For instance, token 1 appears most frequently in answers. Precision = # of times token $i$ is accepted by the target model / # of times token $i$ is proposed by the draft model. Recall = # of times token $i$ is accepted by the target model / # of times token $i$ appears in the final answer.

**High-frequency tokens precision and recall.** In our experiment using the Spider dataset, Vicuna-7M is the target model, and LLaMA-160M is the draft. We identify the top 100 tokens most frequently generated by the target model, which account for 72.2% of all appearances, following a power-law distribution. Figure 5 shows a marked improvement in both accuracy and recall of these tokens after distillation on the test dataset in an offline evaluation.

Table 2: Top 15 tokens with the most recall/precision improvement across datasets. We ignore ⌴ before tokens, which represents space in the LLaMA tokenizer.

| Dataset | Spider | Gsm8k | Alpaca-Finance | Code-Python |
|---|---|---|---|---|
| **Tokens with the greatest precision increase** | AV, SELECT, first, ⟨EOS⟩, template, SUM, G, COUNT, \n, city, WHERE, ';, (, IST, id | ⟨EOS⟩, >>, +, To, <<, this, =, %, know, are, We, calculate, be, The, have | 1, Here, (, :, provide, depends, However, goals, amount, 3, there, The, \n, personal, will | "', (, Here, python, ', how, doc, snippet, import, based, {, Python, This, :, you |
| **Tokens with the greatest recall increase** | SELECT, *, FROM, (, IST, *), \n, COUNT, G, first, WHERE, ⟨EOS⟩, IN, ;, MAX, '; | start, >>, <<, +, find, how, we, =, fore, To, so, \, ⟨EOS⟩, then, let | general, 1, several, This, depends, Here, provide, However, goals, over, (, If, amount, it, can | Here, This, snippet, "', ', how, python, (, takes, Python, you, doc, an, import, def |

**Tokens learned across different datasets** In our study, we analyze the top 10 tokens with the most pronounced accuracy and recall improvements across various datasets, focusing on the 100 most frequent tokens to understand the draft model's learning trends. As detailed in Table 2, the improved tokens align well with the underlying data distribution. For example, in the Spider dataset, which frequently generates SQL statements, tokens like SELECT and WHERE have notably higher acceptance rates post-distillation. These patterns highlight the draft model's ability to adapt and predict tokens consistent with the data distribution.

## 6 CONCLUSION

Speculative decoding's efficiently hinges on the draft model's approximation to the target model. We introduce an online speculative method that continuously enhances the draft model based on varying data distributions. Experiments on both synthetic and real data demonstrate that online speculative decoding swiftly adapts to new data distributions, significantly enhancing token acceptance.

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

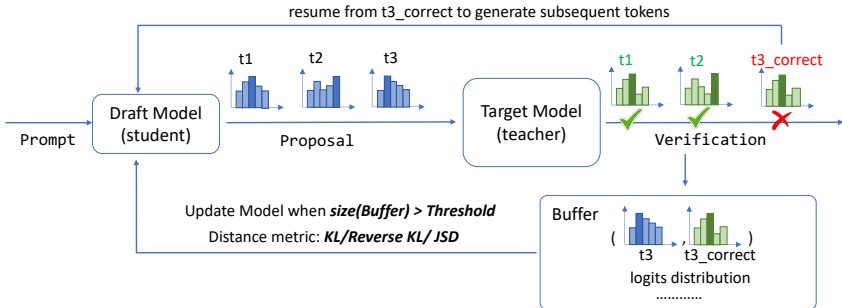

Figure 6: Online speculative decoding overview. For each prompt, the draft model suggests multiple tokens in a single step. The target model then verifies these tokens, accepting some and rejecting others. If the student proposes incorrect tokens, both the draft and target distributions are stored in a buffer. Once the buffer exceeds a specified threshold, the draft model is updated by calculating the loss between the draft and target distributions using various distance metrics.

## A APPENDIX

### A.1 SPEEDUP OF SPECULATIVE DECODING

As proved in Leviathan et al. (2023), compared with standard decoding, the expected improvement factor for offline speculative decoding is $\frac{1-\alpha^{k+1}}{(1-\alpha)(ck+1)}$. Let the time taken for a single run of $M_p$ be $T$. Define $c$, the cost coefficient, as the ratio of the time taken for a single run of $M_q$ to that of $M_p$. Each execution of lines 7 to 8 takes $Tck + T$ and, on average, yields $\frac{1-\alpha^{k+1}}{1-\alpha}$ tokens. As a result, the average time to produce one token using speculative decoding is given by $\frac{(ck+1)(1-\alpha)}{1-\alpha^{k+1}}T$. In contrast, the time to produce a single token using standard decoding is $T$. Hence, the wallclock time reduction of offline speculative decoding can be described as $\frac{1-\alpha^{k+1}}{(1-\alpha)(ck+1)}$.

### A.2 LATENCY ANALYSIS

Suppose OSD can improve the token acceptance rate from $\alpha_1$ to $\alpha_2$ and $T$ is the generation time for standard decoding. Based on Equation 2, this improvement leads to a decrease in the average generation time for each token, transitioning from $\frac{(ck+1)(1-\alpha_1)}{1-\alpha_1^{k+1}}T$ to $\frac{(ck+1)(1-\alpha_2)}{1-\alpha_2^{k+1}}T$. Consequently, this results in a speedup factor of $\frac{1-\alpha_2^{k+1}}{1-\alpha_1^{k+1}}\frac{1-\alpha_1}{1-\alpha_2} = \frac{1+\alpha_2+\alpha_2^2+...+\alpha_2^k}{1+\alpha_1+\alpha_1^2+...+\alpha_1^k}$ compared to standard speculative decoding.

In the aforementioned analysis, we omitted the additional latency due to updating the smaller model for the following reasons: (1) As illustrated subsequently, the additional computational cost (FLOPs) from the update remains marginal when juxtaposed with the computational demands of running the larger model. (2) Updates are periodic, during times of moderate request loads, the latency for serving individual requests remains largely unaffected. Additionally, given that the update operation for the smaller model is considerably less resource-intensive than inference, the associated latency might be seamlessly masked, rendering it virtually imperceptible. Lastly, the processes of updating and inference can even be executed concurrently on separate devices.

### A.3 FLOPS ANALYSIS

*The FLOPs required to update the draft model are significantly fewer than those needed for inference on a large model.* Denote $L$ as the average length of the generated sequence. For each verification, the draft model suggests $k$ tokens. The expected length for a single run of the target LLM, denoted as $a$, can be calculated using Equation 2. Therefore, OSD undergoes the verification process $\frac{L}{a}$ times, with each time verifying $k+1$ tokens. We use $F_{qfwd}$ to represent the arithmetic operations required by a singular forward run of the draft model for each token, and $F_{pfwd}$ stands for the

FLOPs needed for a single forward run of the target model per token. Therefore, the computational demand (in FLOPs) for the draft and teacher models to handle one request can be expressed as: $\text{FLOPs}(draft) = \frac{L}{a} \times k \times F_{qfwd}$, $\text{FLOPs}(target) = \frac{L}{a} \times (k+1) \times F_{pfwd}$. Let's consider the FLOPs required to update the student model per token as $F_{qbwd}$. The cumulative FLOPs necessary to process $I$ requests is given by:

$$\frac{LI}{a} \times [k \times F_{qfwd} + (k+1) \times F_{pfwd}] + I \times L \times F_{qbwd}.$$

Based on the findings of Kaplan et al. (2020), training is approximately three times costlier than inference. This translates to roughly 6 FLOPs per parameter for training on a single token and 2 FLOPs per parameter for inferring on one token. Thus, we can simplify the total FLOPs expression to:

$$\frac{LI}{a} \left[ (k+3a) \times F_{qfwd} + (k+1) \times F_{pfwd} \right]. \tag{6}$$

The proportion of FLOPs needed to run the target model to that of the draft model is given by:

$$\frac{(k+1) \times F_{pfwd}}{(k+3a) \times F_{qfwd}}.$$

For the two model pairs evaluated, assuming an average of 5 proposed tokens per run: (1) (LLaMA-160M, Vicuna7B) with an average acceptance rate of 0.71, the ratio is approximately $\frac{(5+1) \times 7B}{(5+3 \times 3) \times 160M} = 18.75$. (2) (T5-small 80M, Flan-T5-XL 3B), with an average acceptance rate of 0.76, the ratio is roughly $\frac{(5+1) \times 3B}{(5+3 \times 4.3) \times 80M} = 12.6$.

*In practical systems, the FLOPs required for inference are significantly below the machine's capacity.* Consider the LMSYS-Chat-1M Zheng et al. (2023b). It comprises traces spanning 125 days with 1000,000 requests, averaging less than 2,000 tokens per request (including both prompts and responses). When serving a 30B model with 8 A100 GPUs, the FLOPs consumed per second can be estimated as (Still, we estimate 2 FLOPs per token per parameter):

$$\frac{2000 \times 1000,000}{125 \times 24 \times 3600} \times 30 \times 10^9 \times 2 = 5.5 \times 10^9 \text{ FLOPs or 5.5 GFLOPs}$$

On the other hand, 8 A100 GPUs offer a combined capacity of $8 \times 312$ TFLOPs, and the computational utilization is notably low. While Arena (the platform that generates LMSYS-Chat-1M) may not be the most efficient and might lack substantial traffic, it's the only publicly accessible LLM service trace. Even after amplifying the load multiple times, based on the above calculations, the computation efficiency remains limited.

## A.4 BANDWIDTH ANALYSIS

LLM inference is memory bandwidth bound. When the input/output length is short, the memory operations are dominated by loading model parameters from GPU HBM to SRAM. We analyze the memory loading requirements of different inference techniques below ($batch\_size = 1$). We first introduce the notations used in the analysis. $M/m$: The total bytes of the target/draft model. $L$: inference length. $a_1/a_2$: The expected generation length for a single run of the target LLM of Vanilla speculative decoding(VSD)/OSD. $I$: the interval to update the draft model. On a high level, $\frac{L}{a} * M$ represents the bytes required to load the target model, while $L * m$ indicates the bytes needed for loading the draft model. For OSD, $m * \frac{L}{I}$ denotes the bytes necessary to load the draft model for updates.

We applied Formula 2 from our paper to calculate $a_1$, $a_2$, using the token acceptance rates for standard vanilla speculative decoding and OSD on the Spider dataset with the LLaMA-160M and Vicuna-7B models as the draft and target models, respectively. This resulted in $a_1 = 1.4$ and $a_2 = 3.4$. The memory sizes are $M = 14$GB for the target model and $m = 0.32$GB for the draft model. For OSD, the draft model is updated every 8 iterations ($I$=8). Using these values, we have estimated the memory loading bytes, presented in the right column.

As seen in Tabel 3 Updating the draft model is not memory-intensive because the draft model is small. The majority of memory operations are still dominated by loading the target model. (2) OSD can significantly reduce memory bandwidth. This efficiency is achieved through a higher token acceptance rate, which consequently decreases the frequency of calling the larger model for verification.

Table 3: Bandwidth analysis. Original means inference without speculative decoding. VSD, vanilla Speculative Decoding. OSD, online speculative decoding.

| | Memory Loading Formula | Memory Loading in bytes of (LLaMA-160M, Vicuna-7B) pair, $L$=128, $a_1$=1.4, $a_2$=3.4 |
|---|---|---|
| Original | $L * M$ | 1792 GB |
| VSD | $\frac{L}{a_1} * M + L * m$ | 1320 GB |
| OSD | $\frac{L}{a_2} * M + L * m + m * \frac{L}{T}$ | 573 GB |

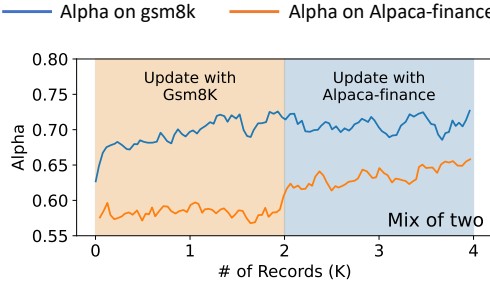

Figure 7: Mix of distributions.

## A.5 DATA MIX

Moreover, there is a question of whether the draft model, once adapted to the new distribution, might lose its prior knowledge. To probe this, we conducted an experiment mixing 2k prompts each from the Gsm8k and Alpaca-finance datasets. During online serving, for the initial 2k requests, we only update the model based on data from the Gsm8k dataset. For the subsequent half of the requests, we restrict updates solely to data from the Alpaca-finance dataset. We then provide the average token acceptance rates for all requests, segmented by their data source (Gsm8k versus Alpaca-finance). As depicted in Figure 7, the token acceptance rate for Gsm8k increases as the draft model is exposed to more data. Conversely, the acceptance rate ($\alpha$) for the Alpaca-finance dataset remains consistent. This is anticipated since we only update the draft model using Gsm8k data. In the latter half of the dataset, the token acceptance rate for the Alpaca-finance dataset also shows an uptrend. Intriguingly, the rate for Gsm8k remains consistent, suggesting that the draft model retains its learned knowledge without showing signs of forgetting.

## A.6 EXPERIMENT SETTING FOR DISTRIBUTION SHIFT ANALYSIS

We employ a single LLaMA-160M as the initial draft model and Vicuna-7B as the target model. To simulate the distribution shift, we integrate data from diverse datasets. Our evaluation focuses on the draft model's token acceptance rate, across varying numbers of data records. To emulate this shift in distribution, we select 2k prompts from each dataset under evaluation. T he data from the four datasets are amalgamated by direct concatenation, such that the records from $i \times 2k$ to $(i+1) \times 2k$ belong solely to dataset $i$.

## A.7 REAL WORKLOADS

**Arena Dataset** For expedited experimental evaluation, we randomly sample a subset with 10K records from LMSYS-Chat-1M Zheng et al. (2023b), a comprehensive real-world LLM conversation dataset. This dataset encompasses interactions with 25 models spanning from April to August 2023 and features conversations in over 150 languages. For all experiments, we only pick conversations for Vicuna models.

**Customize draft models** We propose that employing distinct draft models for queries on various topics can enhance the token acceptance rate. Unlike the approach of utilizing a single draft model for all topics, assigning specific draft models to individual topics narrows the range of query distri-

Table 4: Measured execution time/speedup and theoretical execution time/speedup. Original means inference without speculative decoding.

|  | Original | OSD, $\alpha = 0.5$ | OSD, $\alpha = 0.6$ | OSD, $\alpha = 0.7$ | OSD, $\alpha = 0.8$ | OSD, $\alpha = 0.9$ |
|---|---|---|---|---|---|---|
| Measured time in ms/token (speedup) | 51.09 | 39.90 (1.28 $\times$) | 35.48 (1.44 $\times$) | 30.96 (1.65 $\times$) | 25.42 (2.01 $\times$) | 19.43 (2.63 $\times$) |
| Theoretical time in ms/token (speedup) | 51.09 | 39.00 (1.31 $\times$) | 32.12 (1.59 $\times$) | 26.07 (1.96 $\times$) | 20.77 (2.46 $\times$) | 16.38 (3.12 $\times$) |

butions each model must adapt to. This focused approach simplifies the learning process for each draft model, as they deal with a more limited set of queries. To substantiate this hypothesis, we measured and plotted the token acceptance rates using both strategies - a single universal draft model versus multiple topic-specific draft models - in Figure 8, to highlight the idea of customizing draft model for different types of queries. As seen from the graph, across all topics, employing multiple draft models results in an increase in the token acceptance rate by 0.1 to 0.2. This aligns with our expectation that draft models benefit from a narrower query distribution, making it easier to learn and adapt. We leave it to future research to decide the optimal number of draft models and the best classification strategy.

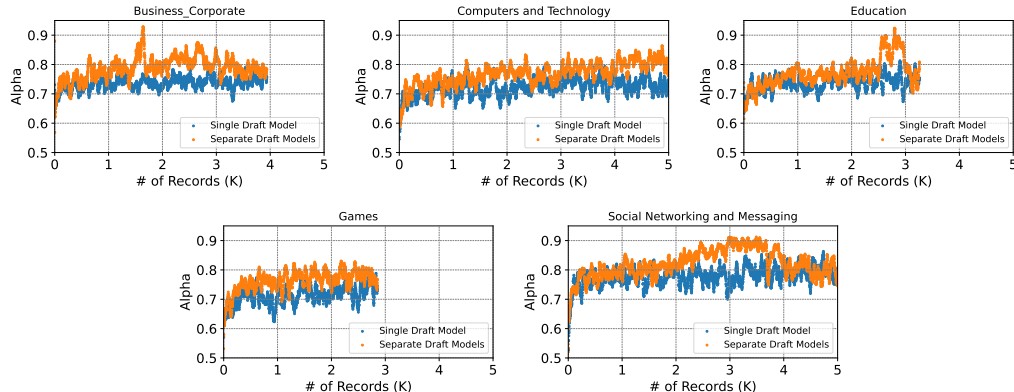

Figure 8: Single draft model vs multiple draft models. For the single draft model, we send all queries to the same draft model and measure the token acceptance rate based on query topics. For multiple draft models, we employ a customized draft model for each query based on the query topic.

## A.8 LATENCY MEASUREMENT

In this section, we evaluate the speedup achieved by OSD. These tests are conducted using llamacpp Gerganov (2023) on an A100 GPU. Initially, we set a constant token acceptance rate to compare the theoretical and actual measured speedups. Following this, we employ OSD distilled model as the draft model and measure the speedup compared with inference without speculative decoding on four distinct datasets. We employ TinyLLaMA-1.1B tin (2023) as the student model and Vicuna 33B as the target model.

From Table 4, OSD can obtain more than 2x speedup in comparison with vanilla LLM inference when we use TinyLLaMA-1.1B as the student model and Vicuna 33B as the teacher model with the above 80% token acceptance rate. Moreover, the observed speedup closely aligns with the theoretical expectations. The primary discrepancies can be attributed to two factors: (1) Slow Sampling: Speculative decoding necessitates additional sampling steps, as the draft model generates preliminary tokens. For optimal performance, the sampling process must be expedited. (2) To attain significant speedup, the execution time ratio (denoted as $c$) between the draft and target models should be minimized. However, in practical implementations, the overall execution time for the draft model is disproportionately affected by kernel launch overheads and Python-related delays, resulting in slower-than-anticipated performance. Lastly, we measure the speedup of OSD

Table 5: (TinyLLaMA-1.1B, Vicuna-33B) measured speedup on four evaluated datasets on a single A100-80G. Inference without speculative decoding has a token latency of 51.09 ms/token.

| Dataset | Spider | Gsm8k | Alpaca-Finance | Code-Python |
|---|---|---|---|---|
| Measured time in ms/token (Speedup) | 23.53 (2.17 ×) | 27.40 (1.89 ×) | 26.53 (1.92 ×) | 30.12 (1.69 ×) |

offline distilled model. We use teacher sampling with forward KD as the distillation method. Using TinyLLaMA-1.1B as the draft model and Vicuna-33B as the target model, we report the token latency and speedup compared with inference without speculative decoding in Table 5. As shown, OSD can achieve 1.64× to 2.09× speedup across four evaluated datasets.

