# OpenReview forum: "Online Speculative Decoding"
_ICLR.cc/2024/Conference — Submitted to ICLR 2024_

### Official Review · Reviewer_fopk · 2023-10-29

**Soundness:** 3 good
**Presentation:** 3 good
**Contribution:** 3 good
**Rating:** 6
**Confidence:** 4

**Summary:**

The author casts the speculative decoding problem as an online knowledge distillation problem. Online speculative decoding works by continually updating draft models on observed user query data using the abundant excess computational power in an LLM serving cluster. This approach frames the learning problem based on the auxiliary information as online knowledge distillation, where the teacher and student models correspond to the target and draft LLMs in speculative decoding, respectively. By doing so, the draft model can be refined in real-time, leading to more effective suggestions and improved predictive accuracy. The benefits of continually updating draft models include more accurate predictions, particularly on data originating from query distributions, and the ability to efficiently and effectively optimize the draft model in real-time.

**Strengths:**

1. Presentation of the idea is clear and straightforward.
2. Evaluation is done thoroughly to understand how online speculative decoding performs under distribution shift to mimic real world scenarios.

**Weaknesses:**

1. In a few places in the paper, the author claims a translation between token acceptance rate and latency reduction. Is this done empirically or theoretically? Throughout the paper, the baseline seems to be against the offline distilled model and how the online model converges and eventually exceeds the performance of the offline distilled model, but the comparison did not include a vanilla model.

2. The author claims an expected improvement over vanilla speculative decoding but does not show it empirically.

3. Fine-tuning would require more computational resources. With more resources, the author could have fitted a larger draft model and performed vanilla speculative decoding. Why do we need an online distilled model in the first place?

4. The author showed the results of the online distilled model after two epochs. What's the performance like during the first two epochs of fine-tuning?

5. If we know that the performance improvement only shows after a certain amount of fine-tuning, does the real-world workload motivate this scenario? It's nice that the author considers the case of distribution shift, but the duration of each phase is also set arbitrarily and does not necessarily reflect the deployment scenario.

Overall, I believe this is a nice paper on online knowledge distillation but the empirical analysis did not capture how it is better than prior speculative decoding approaches. The author proposed a framework that improves the performance of speculative decoding by doing fine-tuning but does not account for how that extra compute can benefit speculative decoding. All comparisons are against other knowledge distillation methods but lack an analysis against known speculative decoding techniques, such as vanilla speculative decoding and tree-based decoding. Also, there are no ablation studies on how hyperparameters in speculative decoding (such as draft model number of new tokens per generation) affect the performance.

**Questions:**

I have listed my questions above.

---

> ### Author Response · Authors · 2023-11-17
>
> Thanks for the review! We address weaknesses and questions below:
>
> **W1.1: Latency reduction calculation:** The reduction in latency is theoretically achieved by applying Equation 2 in the paper. The achievable speedup is decided by the token acceptance rate ($\alpha$) and the execution time ratio ($c$) of running the draft model and the target model. OSD focuses on improving the token acceptance rate and does not affect c. We implemented OSD in llama.cpp. Using LLaMA-1.1B as the draft model  and Vicuna-33B as the target model,  we got the measured and theoretical speedup on A100 below:
>
> | Inference schemes | Baseline w/o speculative decoding | OSD, $\alpha$ = 0.5 | OSD, $\alpha$=0.6 | OSD, $\alpha$=0.7 | OSD, $\alpha$=0.8 | OSD, $\alpha$=0.9|
> |--|--|--|--|--|--|--|
> Measured time in ms/token (speedup) | 51.09 | 39.3 (1.28$\times$) | 35.48 (1.44$\times$) | 30.96 (1.65$\times$) | 25.42 (2.01$\times$) | 19.43 (2.63$\times$) |
> Theoretical time  in ms/token (speedup) | 51.09 | 39.0 (1.31$\times$) | 32.12 (1.59$\times$) | 26.07 (1.96$\times$) | 20.77 (2.46$\times$) | 16.38 (3.12$\times$)|
>
> **W1.2 Vanilla speculative decoding as the baseline:**
> We define vanilla speculative decoding as the method where the unmodified, preliminary draft model is utilized to approximate the target model. **It is essential to note that this approach serves as the baseline for our comparisons.** The token acceptance rates associated with vanilla speculative decoding are listed in the 'Standard' column of Table 1. A significant contribution of this paper is the application of knowledge distillation to speculative decoding, a technique not researched in prior research.
>
> **W1.3 Online speculative decoding baselines:**
> The experiment presented in Section 5.2 is **not** designed to demonstrate that the 'online model outperforms the offline model.' Instead, our objective is to illustrate the performance of the offline model (indicated by the red dashed line in Figure 2) as the maximum potential performance for the online model. This is due to the offline model being constructed assuming **complete awareness of the (future) data distribution** (note OSD does not). Additionally, another baseline we consider is the draft model that has been distilled using only 10% of the data (orange line in Figure 2). The distilled model does not update online. In this context, OSD exhibits superior performance, as it rapidly adapts to and learns from the underlying data distribution.
>
> **W2: Compare with vanilla speculative decoding:** As mentioned above, we **indeed have compared** with the vanilla speculative decoding in the offline setting (vanilla speculative decoding and static model distilled with 10% data) as shown in Table 1 standard column.
>
> **W3: Use a larger draft model:** The choice to utilize a smaller draft model and employ online distillation, as opposed to a larger draft model, is driven by two key considerations (1) Leverage the free FLOPs in model serving:  OSD updates the model parameters only during periods of low load, ensuring that these updates do not compromise serving performance. Additionally, users have the flexibility to dynamically set the interval for updating the draft model.  (2) Balance execution time ratios: As detailed in Equation 2 of the paper, employing a larger draft model increases the execution time ratio between the draft and the target model. This scenario is counterproductive to the goal of speculative decoding. In an extreme case, if the draft model, despite achieving a 100% token acceptance rate, becomes as large as the target model, it negates the latency reduction benefit of speculative decoding. (3) Lastly, we want to use a dynamic draft model because it is necessary to adapt to the online distribution shift.
>
> **W4: Performance of finetuning one epoch:** We report the token acceptance rate $\alpha$ (LLaMA-160M as the draft model and Vicuna-7B as the target model) of the first two epochs in the offline setting below:
> | Dataset | Spider | Gsm8k | Code-search-Python | Alpaca-finance |
> |--|--|--|--|--|
> | $\alpha$ without fine-tuning | 0.28 | 0.58 | 0.38 | 0.57 |
> | $\alpha$ after fine-tuning one epoch | 0.71 | 0.72 | 0.58 | 0.65 |
> | $\alpha$ after fine-tuning two epochs | 0.76 | 0.75 | 0.65 | 0.67 |
>
> **W5: Does the experiment motivate the real deployment scenario:** In Figure 2, we demonstrate that OSD is capable of rapidly increasing the token acceptance rate after observing less than 5K data points. We contend that this is a relatively small sample size in the context of real-world workloads. Regarding Figure 3, the primary aim of this figure is to illustrate OSD's quick adaptability to varying data distributions, even when distribution shift may be arbitrary,  as evident from its analysis of fewer than 2K records. We believe these findings provide robust evidence of OSD's effectiveness in real-world deployments, where the model would be tasked with processing hundreds of thousands of data points.

---

> > ### Comment · Reviewer_fopk · 2023-11-21
> >
> > Thank you for the response and detailed experiments!
> >
> > I have some concerns regarding the responses to W3 and W5. For W3, the author claims that using a larger draft model would be counterproductive. I agree that in the extreme case, it is worse, as otherwise there would be no need for speculative decoding. However, there is a tradeoff space regarding the draft model size, which has not been explored. Also, I am not sure if Equation 2 is verified by empirical experiments, as it seems to have not taken into account context length and batch size, which would vary even for a fixed set of draft models and target models. Similarly for W5, batching would be a strategy to cope with the high-load scenario that the author mentions. The tradeoff space seems to not have been explored.
> >
> > Regarding the newly introduced Figure 6, here the authors fine-tune a different model for a different topic. However, in Figure 4, the performance drop between boundaries seems to indicate that only 1 draft model is deployed and it takes time to catch up. Or are OSD switching models between boundaries in Figure 4? It's not entirely clear what's the deployment setting for Figure 4. If it is the latter case, I am curious how is this classification done.

---

> ### Author Response · Authors · 2023-11-21
>
> Thank you for your response! We greatly value your insightful comments, as they contribute to improving the clarity and quality of our work. We eagerly anticipate further discussions with the reviewers.
>
> **W3: Does not explore the tradeoff between draft model size and accuracy.**
> 1.  To address the reviewer's initial query, which raised concerns about the computational resources required for fine-tuning and the potential for using a larger draft model for vanilla speculative decoding, we have conducted a comparison. Specifically, we have compared the token acceptance rates between fine-tuning LLaMA-160M and utilizing the static TinyLLaMA (1.1B) with vanilla speculative decoding. The results of this comparison across various evaluated datasets are presented below:
>
> |   | Spider | Gsm8k | Code-search-Python | Alpaca-finance |
> |--|--|--|--|--|
> |LLaMA-160M + OSD | 0.76 | 0.75 | 0.65 | 0.67 |
> | TinyLLaMA 1.1B + Vanilla speculative decoding | 0.63 | 0.70 | 0.74 | 0.64 |
>
> As illustrated above, the distilled draft model exhibits higher token acceptance rates on 3 out of the 4 evaluated datasets, even though LLaMA-160M has only 16% of the parameters of the 1.1B model. This underscores the fact that a learned small draft model can deliver comparable or even superior performance compared to a generic large draft model. Additionally, it's worth noting that proposing tokens with the 1.1B model is considerably more resource-intensive than using LLaMA-160M as the draft model.
>
> 2. We would like to emphasize that the primary focus of this project is to highlight the concept of enhancing the draft model online through knowledge distillation. This technique can be broadly applied to draft models of varying sizes. Irrespective of the size of the draft model, OSD ensures a continuous ability to update and enhance the draft model. One of OSD's most significant advantages, unattainable by static models, is its capability to adapt to shifts in data distribution. While exploring the trade-off between the draft model's size and the target model is an important consideration, it is a bit beyond the scope of this paper. For a comprehensive analysis of different model sizes (ranging from 77M to 800M), token acceptance rates, and speed improvements across various datasets, we refer to the original speculative decoding paper [1].
>
> **W5: How does batch size affect speculative decoding?**
> Thank you for your inquiry! Indeed, the batch size might impact the accuracy of Formula 2. This formula operates under the assumption that there are ample computational resources available to facilitate the verification step, as detailed in [1]. This assumption holds most of the time since LLM inference is typically constrained by memory bandwidth or capacity[2][3].  But still, investigating the intersection between speculative decoding and batching requests [2][4] constitutes an intriguing avenue for research. Notably, most existing studies [1][5][6] have concentrated on small batch sizes.  We consider it a potential area for future exploration.
>
> **Clarification for Figure 6 & Figure 4**
>  In Figure 4, we employ a single draft model to showcase OSD's ability to swiftly adapt to distribution shifts. In Figure 6, we demonstrate that tailoring draft models to specific query types yields a higher token acceptance rate compared to utilizing a shared draft model across diverse query types.
>
> *[1] Leviathan, Yaniv, Matan Kalman, and Yossi Matias. "Fast inference from transformers via speculative decoding." International Conference on Machine Learning. PMLR, 2023.*
>
> *[2] Kwon, Woosuk, et al. "Efficient memory management for large language model serving with pagedattention." Proceedings of the 29th Symposium on Operating Systems Principles. 2023.*
>
> *[3] Sheng, Ying, et al. "FlexGen: High-Throughput Generative Inference of Large Language Models with a Single GPU." (2023).*
>
> *[4] Yu, Gyeong-In, et al. "Orca: A distributed serving system for {Transformer-Based} generative models." 16th USENIX Symposium on Operating Systems Design and Implementation (OSDI 22). 2022.*
>
> *[5] https://github.com/FasterDecoding/Medusa*
>
> *[6] Miao, Xupeng, et al. "SpecInfer: Accelerating Generative LLM Serving with Speculative Inference and Token Tree Verification." arXiv preprint arXiv:2305.09781 (2023).*

---

> ### Author Response · Authors · 2023-11-23
> **More results on exploring draft model sizes**
>
> **Model size and token acceptance trade-off**
> | Dataset | Spider | Gsm8k | Code-search-Python | Finance-alpaca |
> | -- | -- | -- | -- | -- |
> | T5-small (60M) + OSD  | 0.78 | 0.62 | 0.81 | 0.63 |
> | T5-base (220M) + OSD | 0.78 | 0.63 | 0.78 | 0.61 |
> | T5-large (770M) + OSD | 0.82 | 0.66 | 0.83 | 0.64 |
>
> The results show T5-small (60M parameters) achieves comparable token acceptance rates in comparison with using a significantly larger T5-large (770M parameters) after fine-tuning, despite T5-small being more than 10 times smaller in size. Hence, we believe that our technique should be applicable to draft models of different sizes.
>
> **More comments on batch size and speculative decoding**
> We would also like to further comment on the remark about batch size and speculative decoding. Speculative decoding is beneficial in systems that are not computationally bounded, which is partially determined by batch size as prior work on continuous batching [1] has shown. Exploring the tradeoff between speculative decoding and continuous batching is challenging, since the decision to speculate depends not only on the batch size, but also on how much computational resources are available and which of the requests from prior batches should be included if any of them failed to verify. We believe this topic merits its own paper and is orthogonal to ODS's contributions, among which is to show the efficacy of online speculation in systems that are not computationally-bounded.
>
> *[1] Yu, Gyeong-In, et al. "Orca: A distributed serving system for {Transformer-Based} generative models." 16th USENIX Symposium on Operating Systems Design and Implementation (OSDI 22). 2022.*

---

### Official Review · Reviewer_rfi3 · 2023-10-31

**Soundness:** 3 good
**Presentation:** 3 good
**Contribution:** 3 good
**Rating:** 6
**Confidence:** 4

**Summary:**

Online speculating decoding proposes the idea of continuously updating the draft model when performing speculative decoding.
The main idea is that "there is spare compute" available when performing auto-regressive decoding. This spare compute can be used to fine-tune the draft model based on the distribution produced LLM.

**Strengths:**

The idea is fairly simple. Continously modifying the draft model can improve the token acceptance rate and provide higher speedups when using speculative decoding.
The authors have explored the space of distillation quite well.

**Weaknesses:**

There are certain points where added clarification of more evaluation will be appropriate. In general I found the evaluation to be underwhelming. Following are specific instances which can be improved.

1. The authors claim there is spare compute as LLM serving is Memory Bandwidth bound. And based on this insight they propose OSD. However, concrete numbers regarding these are missing.Further the evaluation do not talk about runtime, only about token acceptance rates. Here is why I believe this is important, because in my opinion/experiments for most Large LLMs we are on a roofline where we are memomry bandwith bound, even the draft model is going to consume some amount of Memory Bandwidth when performing training. This could adversly effect LLM being served, due to interference. Therefore concrete numbers are going to be useful.


2. My second concern is regarding data mixes. To be fair the authors have done a fair evaluation. However, I believe the evaluation is merely focussed on showing that OSD work. To me to some extent it is straightforward that as a model is fine tuned on the same distribution it starts mimicing, therefore the offline evaluation is kind of straightforward. However, as the authors very well understand (from their online evaluation) it is not very straightforward. I am curious why did the authors decide to have a separate model for each language. Is it a typical scenario for deploying speculative decoding. Further can the authors report speculative decoding numbers on english language without filtering.

3. I would really like to see where the authors think their approach will fail. Are there dataset mixes where this idea will fail. Can we evaluate straight up on LMSys-chat to see how is works without all the filtering.

**Questions:**

Please see the weakness.

---

> ### Author Response · Authors · 2023-11-17
>
> Thanks for the review! We address weaknesses and questions below:
>
> **W1.1 OSD Runtime numbers:**
> | Inference schemes | Baseline w/o speculative decoding | OSD, $\alpha$ = 0.5 | OSD, $\alpha$=0.6 | OSD, $\alpha$=0.7 | OSD, $\alpha$=0.8 | OSD, $\alpha$=0.9|
> |--|--|--|--|--|--|--|
> |Measured time in ms/token (speedup) | 51.09 | 39.9 (1.28$\times$) | 35.48 (1.44$\times$) | 30.96 (1.65$\times$) | 25.42 (2.01$\times$) | 19.43 (2.63$\times$) |
> |Theoretical time  in ms/token (speedup) | 51.09 | 39.0 (1.31$\times$) | 32.12 (1.59$\times$) | 26.07 (1.96$\times$) | 20.77 (2.46$\times$) | 16.38 (3.12$\times$)|
>
> Moreover, the observed speedup closely aligns with the theoretical expectations. The primary discrepancies can be attributed to two factors: (1) Slow Sampling: Speculative decoding needs additional sampling steps, as the draft model generates preliminary tokens. For optimal performance, the sampling process must be expedited. (2) To attain significant speedup, the execution time ratio (denoted as 'c') between the draft and target models should be minimized. However, in practical implementations, the overall execution time for the draft model is disproportionately affected by kernel launch overheads and Python-related delays, resulting in slower-than-anticipated performance. We are actively working on solving those engineering problems.
>
> **W1.2 OSD might consume more memory bandwidth because of updating the draft model:**
> As pointed out by the reviewer, LLM inference is memory bandwidth bound. When the input/output length is short, the memory operations are dominated by loading model parameters from GPU HBM to SRAM. We analyze the memory loading requirements of different inference techniques below (batch_size=1).
>
> We first introduce the notations used in the analysis. $M$/$m$: The total bytes of the target/draft model. $L$: inference length. $a_1$/$a_2$: The expected generation length for a single run of the target LLM of Vanilla speculative decoding(VSD)/OSD. $I$:  the interval to update the draft model.
>
> On a high level, $\frac{L}{a} * M$ represents the bytes required to load the target model, while $L * m$ indicates the bytes needed for loading the draft model. For OSD, $m * \frac{L}{I}$ denotes the bytes necessary to load the draft model for updates.
>
> |     | Memory Loading Formula | Memory Loading bytes of (LLaMA-160M, Vicuna-7B) pair, $L$=128, $a_1$=1.4, $a_2$=3.4 |
> | -- | -- | -- |
> | Inference without speculative decoding | $L * M$ | 1792 GB|
> | Vanilla speculative decoding (VSD) | $\frac{L}{a_1} * M + L * m $ | 1320 GB |
> | OSD | $\frac{L}{a_2} * M + L * m + m * \frac{L}{I} $ | 573 GB |
>
> We applied Formula 2 from our paper to calculate $a_1$, $a_2$, using the token acceptance rates for VSD and OSD on the Spider dataset with the LLaMA-160M and Vicuna-7B models as the draft and target models, respectively.
> This resulted in $a_1 = 1.4$ and $a_2 = 3.4$. The memory sizes are $M$ = 14GB for the target model and $m$ = 0.32GB for the draft model. For OSD, the draft model is updated every 8 iterations ($I$=8). Using these values, we have estimated the memory loading bytes, presented in the right column.
>
> As seen in table (1) Updating the draft model is not memory-intensive because the draft model is small. The majority of memory operations are still dominated by loading the target model. (2) OSD can significantly reduce memory bandwidth. This efficiency is achieved through a higher token acceptance rate, which consequently decreases the frequency of calling the larger model for verification.
>
> **W2.1:** Could you explain “However, as the authors very well understand (from their online evaluation) it is not very straightforward.”?
>
> **W2.2: Why use multiple draft models?** We propose that employing distinct draft models for queries on various topics can enhance the token acceptance rate. Unlike utilizing a single draft model for all topics, assigning specific draft models to individual topics narrows the range of query distributions each model must adapt to. This focused approach simplifies the learning process for each draft model, as they deal with a more limited set of queries.
> To substantiate this hypothesis, we measured and plotted the token acceptance rates using both strategies - a single universal draft model versus multiple topic-specific draft models in Figure 6 in the revised paper, to highlight the idea of customizing the draft model for different types of queries. As seen from the graph, across all topics, employing multiple draft models results in an increase in the token acceptance rate by 0.1 to 0.2. This aligns with our expectation that draft models benefit from a narrower query distribution, making it easier to learn and adapt. We leave it to future research to decide the optimal number of draft models and the best classification strategy.
>
> **Please check the revised paper for Figure 6, please download the PDF instead of using the browser to open the PDF to avoid rendering issues. Thanks!**

---

> > ### Author Response · Authors · 2023-11-17
> >
> > **W3: When will OSD fail?** The effectiveness of OSD can be impacted
> > (1) **Rapid Changes in Query Distribution:** OSD may struggle in situations where the query distribution is in flux. If the distribution changes too quickly, the draft model faces challenges in adapting to such a diverse data range. This situation can lead to a stagnation or decrease in the token acceptance rate, diminishing the efficacy of OSD.
> > (2) **Limitations in Compute-Bound Systems:** In systems that are already compute-bound, the trade-off that speculative decoding offers—extra computation in exchange for reduced memory operations—may not be beneficial. This is particularly true in batch-serving systems with very large batch sizes under high request loads, where the system's computational capacity is already fully utilized, rendering speculative decoding less effective. This constraint is more broadly a limitation of speculative decoding in general, rather than being specific to OSD.

---

> > > ### Author Response · Authors · 2023-11-21
> > >
> > > Dear reviewer rfi3, we kindly encourage our reviewers to provide additional comments and clarifications. We are more than willing to respond to any inquiries and address any feedback. Thank you very much!

---

> > > > ### Author Response · Authors · 2023-11-23
> > > > **More measured performance of OSD**
> > > >
> > > > Based on reviewers' requests, we measure  OSD offline distilled model on a single A100-80G across four evaluated datasets in the paper. We use teacher sampling with forward KD as the distillation method. Using TinyLLaMA-1.1B as the draft model and Vicuna-33B as the target model, we report the token latency and speedup compared with inference without speculative decoding in the table below. Inference without speculative decoding has a token latency of 51.09 ms/token and OSD can achieve 1.69$\times$ to 2.17$\times$ speedup across four evaluated datasets.
> > > >
> > > > | Dataset | Spider | Gsm8k | Alpaca-Finance | Code-Python |
> > > > | -- | -- | -- | -- | -- |
> > > > |Measured time in ms/token (speedup) | 23.53 (2.17 $\times$) | 27.40 (1.89 $\times$) | 26.53 (1.92 $\times$) | 30.12 (1.69 $\times$)|

---

> > > > > ### Author Response · Authors · 2023-11-23
> > > > >
> > > > > Dear reviewer rfi3, it's almost near the end of the discussion session, we encourage our reviewers to provide additional comments and clarifications. We are more than willing to respond to any inquiries and address any feedback. Thanks!

---

### Official Review · Reviewer_g3SB · 2023-11-01

**Soundness:** 3 good
**Presentation:** 3 good
**Contribution:** 2 fair
**Rating:** 6
**Confidence:** 2

**Summary:**

Distilling LLM to smaller models for effective online performance is an active area of research and authors focus on this and propose an online speculative decoding approach to effectively perform this.
They use knowledge distillation using KL divergence loss and train a smaller model from teacher model.
They show that their model outperforms static FLAN-T5 in performance.

**Strengths:**

Shows that the online decoding (i am assuming trianing as well) helps improve acceptance rate compared to offline static training.

**Weaknesses:**

A bit hard to understand the novelty and contribution.
Experiment baselines seem a bit lacking.

**Questions:**

I am may have missed somethings, but below are some of my questions.
It is unclear on what the true novelty of the paper is. If i understand correctly you are performing online decoding and training of draft model to adapt to distribution shift.
Also during the online distribution shift evaluation you do a sequential evaluation, what happens when you mix the data and evaluate? and what is the performance of static model on the same?

---

> ### Author Response · Authors · 2023-11-18
>
> Thanks for the review! We address weaknesses and questions below:
>
> **W1:   A bit hard to understand the novelty and contribution.**
>
> OSD  introduces novelty in several key aspects:
> (1) **Online Setting:** Unlike previous speculative decoding research, which predominantly focused on offline settings with a static draft model, OSD operates in an online environment. This method is better aligned with real-world situations, where open-domain draft models often exhibit limited speculation accuracy and query distributions are subject to fluctuation. Consequently, OSD offers a solution that is both more adaptable and practically applicable in dynamic environments.
> (2) **New Techniques in Knowledge Distillation:** OSD is the first to employ knowledge distillation as a method to update the draft model. Knowledge distillation can significantly increase the token acceptance rate compared with traditional finetuning on T5 models across different tasks.
> (3) **Validation and Analysis in Real-World Contexts:** OSD has been proven effective in real-world workloads, specifically using the real-world arena dataset. In addition, we provide an analysis of real-world settings, arguing that the abundance of free Floating Point Operations (FLOPs) in many systems means that updating a small model online incurs negligible overhead. This underlines OSD's practical viability and efficiency.
>
> **W2:    during the online distribution shift evaluation you do a sequential evaluation, what happens when you mix the data and evaluate?**
>
> Could you clarify "Also during the online distribution shift evaluation you do a sequential evaluation, what happens when you mix the data and evaluate? and what is the performance of static model on the same?" Are you asking the baseline that are distilled from the mix dataset? Or are you interested in OSD's performance on mixed dataset?
>
> In the experiment shown in Figure 4, the goal is to evaluate OSD’s effectiveness during the online distribution shift. The aim was to demonstrate that OSD can rapidly adjust to new distributions, eventually aligning with the performance of offline distillation. To establish our baselines for this evaluation, we used offline models distilled with varying proportions of the total training data (30%, 50%, 70%, and 100%) from each dataset.
> Empirically, we anticipated that our baselines, being tailored to specific downstream tasks, would outperform those distilled from a mixed data set. Our focus was not on the mixed data set in evaluating our online method, as our primary interest lies in serving scenarios where a user's query distribution might transition between distinct tasks upon completion.

---

> ### Author Response · Authors · 2023-11-21
>
> Dear reviewer g3SB, we kindly encourage our reviewers to provide additional comments and clarifications. We are more than willing to respond to any inquiries and address any feedback. Thank you very much!

---

> > ### Author Response · Authors · 2023-11-23
> >
> > Dear reviewer g3SB, it's almost near the end of the discussion session, we encourage our reviewers to provide additional comments and clarifications. We are more than willing to respond to any inquiries and address any feedback. Thanks!

---

> ### Comment · Reviewer_g3SB · 2023-11-23
> **Thanks for the response.**
>
> thanks for the clarification of novelty.
>
> Question about experiment was on what is the performance on static dataset where have train and test data by randomly sampling (not online setting). I understand the primary focus of the paper is on online approach, but it would be good to have the other performance as well.
> Also i am not sure if it is right to call the experiments distribution shift as they are different datasets all together, generally distribution shift happens over a period of time in the realworld. In the experiment here its complete change of task rather than a gradual shift right?
>
> Thanks

---

> ### Author Response · Authors · 2023-11-23
>
> Thank you for your follow-up and clarification.
>
> We provided results that demonstrate our method's effectiveness (improvement in token acceptance rates) for the offline setting in Table 1. In addition, to address your concerns, we have performed additional experiments (Vicuna-7B as teacher, LLaMA-160M as student) that compare tailoring draft models to specific query types versus utilizing a shared draft model across diverse query types:
>
> | draft model type | education | game | social | computer |
> |:---:|:---:|:---:|:---:|:---:|
> | specialized model | 0.80 | 0.77 | 0.85 | 0.82 |
> | shared model | 0.74 | 0.73 | 0.78 | 0.72 |
>
> Specialized model refers to student models distilled from a single query type, while shared model refers to student model distilled from a mixture of all four query types (from the Chatbot Arena dataset). As seen from the table, across all topics, employing multiple draft models results in an increase in the token acceptance rate by a noticeable margin (~10%). This aligns with our expectation that draft models benefit from a narrower query distribution, making it easier to learn and adapt.
>
> Regarding distribution shifts during online learning, we indeed provided empirical evidence that OSD works well in the cases where user queries suddenly shift from one topic to another totally irrelevant one (Figure 3). In cases of minor distribution shifts, drop in token acceptance rate is even smaller and OSD is able to easily catch up with the performance drop as long as there is no persistent and rapid changes in query distribution. When facing more complicated and heterogenous query distribution, one strategy is to assign tailored draft models to individual topics and narrow the range of query distribution each model adapts to. OSD is proven to be effective in this use case (Figure 4). We leave it to future research to decide the optimal number of draft models and the best classification strategy.
>
> Please consider improve the rating if we successfully addressed your concerns. Thanks!
>
> Best,
>
> Authors

---

### Official Review · Reviewer_y71B · 2023-11-06

**Soundness:** 3 good
**Presentation:** 3 good
**Contribution:** 3 good
**Rating:** 6
**Confidence:** 3

**Summary:**

This paper proposes online speculative decoding, which utilizes online knowledge distillation to update the small draft model, to improve the acceptance rate. The results show a substantial increase in the token acceptance rate by 0.1 to 0.48, which translates into 1.22x to 2.42x latency reduction.

**Strengths:**

1. This work is the first one that introduces the online draft model update to speculative decoding models, while previous speculative decoding models all assume a static draft model.
2. This paper provides a thorough theoretical analysis to evaluate the speedup, latency, and flops.

**Weaknesses:**

1. Lack of comparison with SOTA works using "multiple draft models". One example [1].
2. The speedup is theoretically estimated. Lack of real-hardware evaluation.

[1] https://github.com/FasterDecoding/Medusa

**Questions:**

1. Could the authors compare the proposed online speculative decoding to the multi-head speculative decoding work [1]? For example, can the proposed online update [1]? What are the potential challenges?
2. Could the authors show real hardware evaluation results?

[1] https://github.com/FasterDecoding/Medusa

---

> ### Author Response · Authors · 2023-11-17
>
> Thanks for the review! We address weaknesses and questions below:
>
> **W1: Could the authors compare the proposed online speculative decoding to the multi-head speculative decoding work?**
> Thank you for mentioning Medusa. The main differences between OSD and Medusa are as follows:
> (1) Medusa needs extra training on a large model: The training process for Medusa requires more resources. Despite Medusa only fine-tuning the additional heads while keeping the original target model frozen, it still needs to run the forward pass on the entire target model. This aspect of the training process remains costly. Empirically, we compare the training time of Medusa heads of Vicuna-7B and the OSD distillation cost on LLaMA-160M as the draft model for Vicuna-7B in the following table:
>
> |   | Extra parameters | Offline training time on a single A100 GPU for 1 epoch |
> |--|--|--|
> | Medusa | 0.44B | 7h |
> | OSD | 0.16B | 32min |
>
> (2) Medusa, being an offline algorithm, cannot adapt to shifts in data distribution, unlike the online nature of OSD: It requires pre-known target distributions and specific datasets to fine-tune its components. During distribution shifts, Medusa's static weights may lead to reduced token acceptance rates. Conversely, OSD is online and adapts quickly and automatically to distribution changes by periodically updating the draft model during non-peak hours, ensuring more responsive and efficient adjustments.
>
> (3) Lastly, at this point, Medusa only supports decoder-only generative models like LLaMA while OSD is generally applicable to all transformer models.
>
> **W2: For example, can the proposed online update [1]? What are the potential challenges?**
> Certainly, OSD can be integrated into Medusa for online serving with concurrent model head updates. However, this approach presents certain challenges. Notably, training the Medusa heads demands more FLOPs compared to training the draft model, as estimated previously. Therefore, implementing Medusa with OSD would be more feasible when the system is under a lower load.
>
> **Q2: The speedup is theoretically estimated. Lack of real-hardware evaluation. Could the authors show real hardware evaluation results?**
> In comparison with vanilla LLM inference, we can achieve measured speedup using our online distilled draft models developed on top of the llama.cpp repo [2]. The experiments are conducted on a single A100 GPU with batch size = 1 and draft token length = 8,  results are attached below:
> | Inference schemes | Baseline w/o speculative decoding | OSD, $\alpha$ = 0.5 | OSD, $\alpha$=0.6 | OSD, $\alpha$=0.7 | OSD, $\alpha$=0.8 | OSD, $\alpha$=0.9|
> |--|--|--|--|--|--|--|
> |Measured time in ms/token (speedup) | 51.09 | 39.9 (1.28$\times$) | 35.48 (1.44$\times$) | 30.96 (1.65$\times$) | 25.42 (2.01$\times$) | 19.43 (2.63$\times$) |
> |Theoretical time  in ms/token (speedup) | 51.09 | 39.0 (1.31$\times$) | 32.12 (1.59$\times$) | 26.07 (1.96$\times$) | 20.77 (2.46$\times$) | 16.38 (3.12$\times$)|
>
> From the table, OSD can obtain more than 2x speedup in comparison with vanilla LLM inference when we use TinyLLaMA-1.1B [3] as the student model and Vicuna 33B as the teacher model with an above 80% token acceptance rate. Moreover, the observed speedup closely aligns with the theoretical expectations. The primary discrepancies can be attributed to two factors: (1) Slow Sampling: Speculative decoding necessitates additional sampling steps, as the draft model generates preliminary tokens. For optimal performance, the sampling process must be expedited. (2) To attain significant speedup, the execution time ratio (denoted as 'c') between the draft and target models should be minimized. However, in practical implementations, the overall execution time for the draft model is disproportionately affected by kernel launch overheads and Python-related delays, resulting in slower-than-anticipated performance. We are actively working on solving those engineering problems.
>
> [1] https://github.com/FasterDecoding/Medusa
>
> [2] https://github.com/ggerganov/llama.cpp
>
> [3] https://github.com/jzhang38/TinyLlama

---

> > ### Author Response · Authors · 2023-11-21
> >
> > Dear reviewer y71B, we kindly encourage our reviewers to provide additional comments and clarifications. We are more than willing to respond to any inquiries and address any feedback. Thank you very much!

---

> > > ### Author Response · Authors · 2023-11-23
> > > **More measured performance of OSD**
> > >
> > > Based on reviewers' requests, we measure  OSD offline distilled model on a single A100-80G across four evaluated datasets in the paper. We use teacher sampling with forward KD as the distillation method. Using TinyLLaMA-1.1B as the draft model and Vicuna-33B as the target model, we report the token latency and speedup compared with inference without speculative decoding in the table below. Inference without speculative decoding has a token latency of 51.09 ms/token and OSD can achieve 1.69$\times$ to 2.17$\times$ speedup across four evaluated datasets.
> > >
> > > | Dataset | Spider | Gsm8k | Alpaca-Finance | Code-Python |
> > > | -- | -- | -- | -- | -- |
> > > |Measured time in ms/token (speedup) | 23.53 (2.17 $\times$) | 27.40 (1.89 $\times$) | 26.53 (1.92 $\times$) | 30.12 (1.69 $\times$)|

---

> > > > ### Author Response · Authors · 2023-11-23
> > > >
> > > > Dear reviewer y71B, it's almost near the end of the discussion session, we encourage our reviewers to provide additional comments and clarifications. We are more than willing to respond to any inquiries and address any feedback. Thanks!

---

### Author Response · Authors · 2023-11-23
**Rebuttal Summary**

We would like to express our gratitude to all reviewers for their feedback. In this post, we aim to:

1. Highlight the positive aspects noted in the reviews.
2. Outline the modifications made in response to the reviewers' inquiries.

**Key Positive Aspects:**
1. [y71B] This is the first work that discusses speculative decoding in the online setting.
2. [y71B] This paper provides a thorough theoretical analysis to evaluate the speedup, latency, and flops.
3. [rfi3] Thoroughly explore the space of applying knowledge distillation for the draft model.
4. [fopk] Evaluation is done thoroughly to understand how online speculative decoding performs under distribution shift to mimic real-world scenarios.
5. [fopk] Good presentation and clear idea.

**Actions Taken During the Rebuttal Phase:**
1. [y71B, rfi3, fopk] We have included latency measurements for the Vicuna-33B model on real A100 hardware in Appendix A.8.
2. [rfi3, fopk] We demonstrate the necessity for customized draft models by contrasting the token acceptance rates between using individual draft models and a singular draft model, as detailed in Appendix A.7.
3. [rfi3] We have incorporated a bandwidth analysis of Online Speculative Decoding (OSD) in Appendix A.4.
4. [y71B] We elucidate the relationship and differences between OSD and Medusa.
5. [fopk] We compare the effectiveness, in terms of token acceptance rate, between different tailored smaller draft models and the more generalized larger draft models.

---

### Meta-Review · Area_Chair_zc9j · 2023-12-10

**Metareview:**

The paper describes an idea to update the draft model adaptively during serving using knowledge distillation techniques so that the draft model more easily adapt to the prompt distributions. Overall, I think this paper is borderline, and I am leaning to not recommending the paper.

Strength:
1. The idea is presented clearly and results look promising.
2. It could be beneficial to query distributions that are changing fast to maximize speedup.

Weakness:
1. The novelty of the paper is limited. Changing the loss function of a draft model using distillation in an offline setting can also be straightforward and it might be easier to do so as it is offline.
2. Actually deployment could be challenging to support online updating. In many of the deployment environments, only inference is supported due to speed / latency / throughput constraints. To actually allow a model being updated (even it is a draft model), a separate environment to support training would be likely needed, syncing the models between these environments in a certain way. And it is unclear whether it is worth the effort.

**Justification For Why Not Higher Score:**

An interesting idea for inference speedup, but its novelty is limited, and unlikely to be practical.

**Justification For Why Not Lower Score:**

None

---

### Decision · Program_Chairs · 2024-01-16

Reject